# New Andean source of resistance to anthracnose and angular leaf spot: Fine-mapping of disease-resistance genes in California Dark Red Kidney common bean cultivar

**M. C. Gonçalves-Vidigal**[1]*, **T. A. S. Gilio**[1], **G. Valentini**[1], **M. Vaz-Bisneta**[1], **P. S. Vidigal Filho**[1], **Q. Song**[2], **P. R. Oblessuc**[3], **M. Melotto**[3]

**1** Departamento de Agronomia, Universidade Estadual de Maringá, Av. Colombo, Maringá, Paraná, Brazil, **2** Soybean Genomics and Improvement Laboratory, USDA-ARS, BARC-West, Beltsville, Maryland, United States of America, **3** Department of Plant Sciences, University of California, Davis, California, United States of America

\* mcgvidigal@uem.br

## Abstract

Anthracnose (ANT) and angular leaf spot (ALS) caused by *Colletotrichum lindemuthianum* and *Pseudocercospora griseola*, respectively, are devastating diseases of common bean around the world. Therefore, breeders are constantly searching for new genes with broad-spectrum resistance against ANT and ALS. This study aimed to characterize the genetic resistance of California Dark Red Kidney (CDRK) to *C. lindemuthianum* races 73, 2047, and 3481 and *P. griseola* race 63–39 through inheritance, allelism testing, and molecular analyses. Genetic analysis of response to ANT and ALS in recombinant inbred lines (RILs) from a CDRK × Yolano cross (CY) showed that the resistance of CDRK cultivar is conferred by a single dominant loci, which we named $CoPv01^{CDRK}$/$PhgPv01^{CDRK}$. Allelism tests performed with race 3481showed that the resistance gene in CDRK is independent of the *Co-1* and *Co-AC*. We conducted co-segregation analysis in genotypes of 110 CY RILs and phenotypes of the RILs in response to different races of the ANT and ALS pathogens. The results revealed that $CoPv01^{CDRK}$ and $PhgPv01^{CDRK}$ are coinherited, conferring resistance to all races. Genetic mapping of the CY population placed the $CoPv01^{CDRK}$/$PhgPv01^{CDRK}$ loci in a 245 Kb genomic region at the end of Pv01. By genotyping 19 RILs from the CY population using three additional markers, we fine-mapped the $CoPv01^{CDRK}$/$PhgPv01^{CDRK}$ loci to a smaller genomic region of 33 Kb. This 33 Kb region harbors five predicted genes based on the common bean reference genome. These results can be applied in breeding programs to develop bean cultivars with ANT and ALS resistance using marker-assisted selection.

## Introduction

*Phaseolus vulgaris* L. (common bean) is the most commonly consumed *Phaseolus* species worldwide [1, 2, 3], and it is an important primary source of protein in several countries. In

**Data Availability Statement:** All relevant data are within the paper and its Supporting Information files.

**Funding:** This research was supported by Brazilian Federal Funding Institutions National Council for Scientific and Technological Development (CNPq) for financial support and scholarship grants and the Coordination for the Improvement of Higher Education Personnel (Capes). M.C. Gonçalves-Vidigal is grateful for grant from Capes number BEX 88881.170662//2018-01. The funders had no role in study design, data collection and analysis, decision to publish, or preparation of the manuscript.

**Competing interests:** The authors declare that they have no conflicts of interest.

**Abbreviations:** ARS, Agricultural Research Service; NB-LRR, nucleotide-binding leucine-rich repeat; Nupagri, Núcleo de Pesquisa Aplicada à Agricultura; PTK, protein tyrosine kinase; SNP, single nucleotide polymorphism; STK, serine-threonine protein kinase; TNL, N-terminal *Toll*-interleukin-1 receptor (TIR)-like domain; UEM, Universidade Estadual de Maringá.

particular, common beans are consumed in large quantities in many areas of Africa and Latin America and are part of traditional diets in the Middle East and Europe [4, 5]. However, this legume is susceptible to several diseases threatening its production worldwide [6]. For instance, anthracnose (ANT), caused by *Colletotrichum lindemuthianum* (Sacc. and Magnus) Briosi and Cavara, and angular leaf spot (ALS), caused by *Pseudocercospora griseola* (Sacc.) Crous and Braun, are the most widespread, recurrent, and devastating diseases of the common bean in Latin America and Africa [7, 8, 9, 10, 11]. Under favorable climatic conditions for the pathogen and with the use of susceptible cultivars or infected seeds, ANT and ALS can cause field losses of up to 100% and 80%, respectively [7, 12, 13, 14]. Importantly, ANT is not limited to the tropics and is one of the major diseases of beans throughout temperate regions as well [15, 16].

The most desirable strategy to control ANT and ALS is the use of resistant cultivars, which can reduce yield losses without the negative environmental impact of fungicide application [17, 18, 19]. However, the implementation of resistance is challenged by the recurrent emergence of virulence phenotypes in the pathogen population, usually referred to races of *C. lindemuthianum* and *P. griseola*. New virulence races of these pathogens have resulted in reduced or complete loss of yield in previously resistant commercial cultivars [17, 20, 21, 22]. Thus, new sources of durable resistance are highly desirable for effective breeding efforts. Gene pyramiding for durable resistance to diseases caused by highly variable pathogens is greatly facilitated by marker-assisted selection, and the development of highly accurate molecular markers that are tightly linked to important disease-resistance genes enables the pyramiding of these genes into single cultivars with broad-spectrum resistance.

Resistance to ANT is conferred by independently segregating individual loci in a series named *Co*. Currently, the known *Co* genes are *Co-1* and its alleles, *Co-14*, *Co-Pa*, *Co-x*, and *Co-w* on chromosome Pv01 [22, 23, 25, 27, 28, 29, 30, 31]; *Co-u* and *CoPv02* on chromosome Pv02 [18, 22]; *Co-13* and *Co-17* on chromosome Pv03 [32]; *Co-3*, *Co-3²*, *Co-3³*, *Co-3⁴/Phg-3*, *Co-y*, *Co-z*, and *Co-RVI* on chromosome Pv04 [24, 26]; *Co-5*, *Co-6*, and *Co-v* on chromosome Pv07; and *Co-2* on chromosome Pv11 [32].

Although resistance to the ALS pathogen is largely conferred by single dominant resistance loci identified by classical genetic approaches, quantitative trait loci (QTLs) have recently been found as well [33, 34, 35, 36]. To date, five resistance loci (the *Phg* series) have been mapped to the integrated bean linkage map; the three independent loci *Phg-1*, *Phg-2*, and *Phg-3* are located on chromosomes Pv01, Pv08, and Pv04, respectively [25, 26, 37], while the two major QTLs *Phg-4* and *Phg-5* are located on Pv04 and Pv10 [33, 34, 36, 38, 39, 40].

Importantly, these *Co* and *Phg* loci may be part of disease-resistance clusters on various chromosomes [23, 24, 25, 28, 32, 41]. For instance, linkage group Pv01 contains a gene cluster having ANT (*Co-1*, *Co-AC*, *Co-14*, *Co-x*, and *Co-w*), rust (*Ur-9*), and ALS (*Phg-1*) resistance genes [18, 25, 26, 28, 32, 29]. The major resistance genes for bean rust, ANT, common bacterial blight, and white mold were mapped in clusters on chromosomes Pv01, Pv04, Pv07, and Pv11 [42, 43]. In addition, a set of resistance gene analogs (RGAs) that are linked to resistance loci for different common bean pathogens were identified [44]. These observations highlight the opportunity for the simultaneous selection of resistance to multiple diseases, especially diseases that occur in similar geographic regions, such as ANT and ALS. The Andean common bean cultivar California Dark Red Kidney (CDRK) is resistant to Mesoamerican races of ANT reported in Brazil, Argentina, and Colombia; all of the Central American races; and races present in Africa and Europe [45, 46, 47, 48, 49, 50, 51].

The availability of reference genomes for the Andean G19833 [41] and Mesoamerican BAT93 [52] has enabled the fine-mapping of many resistance loci, including *Co-AC*, Co-1^HY, *Co-x*, and *Co-1^73-X* on Pv01 [28, 29, 31, 53] and *Co-4²* on Pv08 [54]. In this study, we verified

the resistance inheritance in CDRK. We also found that CDRK is an excellent source to test the hypothesis that ANT and ALS resistance are colocalized in the bean genome and to fine-map the resistance loci using the recombinant inbred line (RIL) mapping population California Dark Red Kidney × Yolano (CY), molecular markers and reference genome.

## Material and methods

### Plant material and growth conditions

To determine the genetic basis of disease resistance in the genotype CDRK, we used 110 RILs derived from the CY population described by [55]. Seeds were sown in plastic trays (50 × 30 × 9 cm) containing a mixture of the commercial substrate MecPlant (Register EP PB 09549-4/ Mapa Brazil, MEC PREC—Ind. Com Ltda, Telemaco Borba, PR), which had been previously fertilized and sterilized. Seedlings were grown under natural light in greenhouses supplemented with 400 W high-pressure sodium lamps, providing a total light intensity of 115 μmoles $m^{-2}$ $s^{-1}$ for 15 days until the seedlings reached the first fully expanded trifoliate leaf stage V3 [56]. Plants were inoculated at this stage.

Additionally, cultivars including Michigan Dark Red Kidney (MDRK), Kaboon, Perry Marrow, AND 277, Widusa, Jalo Vermelho, Jalo Listras Pretas, Pitanga, Corinthiano, Paloma, Amendoim Cavalo, Jalo Pintado 2 and Jalo EEP 558 were evaluated for ANT resistance as well as ALS resistance to race 63–39. The experiments comply with the current laws of the country in which they were performed.

### Pathogenesis assays

To determine the spectrum of resistance in the above bean genotypes, seedlings were inoculated with the Mesoamerican races 9, 64, 65, 73, 89, 445, 453, 1545, 2047, and 3481, as well as the Andean races 2, 7, 19, 23, 39, and 55 of *C. lindemuthianum*. Furthermore, seedlings of the CY RIL population were inoculated with races 73, 2047, and 3481 of *C. lindemuthianum* and race 63–39 of *P. griseola* to determine segregation patterns of the disease reaction in the CY RIL population.

Monosporic cultures of *C. lindemuthianum* and of *P. griseola* were prepared according to the methodologies proposed by Mathur et al. [57] and Sanglard et al. [58]. Inoculum of the ANT races were produced on green common bean pod medium [60] incubated at 22°C for 14 days. The inoculum of race 63–39 of ALS was multiplied in Petri dishes containing 2 mL tomato medium [58] and maintained in a BOD incubator at 24°C for 18 days.

Soon after the expansion of the first trifoliolate leaf, the leaflet of 10 seedlings of each line were inoculated with each race of *C. lindemuthianum* and the leaflet of 10 seedlings with *P. griseola*. Each pathogen was inoculated separately. A spore suspension containing $2.0 \times 10^6$ spores $mL^{-1}$ of races 73, 2047 and 3481 of *C. lindemuthianum* were inoculated using a DeVilbiss number 15 atomizer powered by an electric air compressor (Schulz, SA, Joinville, Santa Catarina, Brazil). A similar procedure was employed for the inoculation with race 63–39 of *P. griseola* using a spore suspension of $1.2 \times 10^4$ conidia $mL^{-1}$. Ten plants for resistant and susceptible control for each race were inoculated [59, 60].

After inoculation, the plants were maintained at >95% relative humidity, 20 ± 2°C, and 12 h of daily light (680 lux) in a mist chamber for three days. The plants were then transferred to open benches under the same conditions, except for the high humidity, for 7 days (ANT) and 17 days (ALS). ANT and ALS symptoms were evaluated using the disease severity scales (1 to 9) proposed by Pastor-Corrales et al. [61] and Inglis et al. [62]. Plants with disease reaction scores from 1 and 3 were considered resistant, whereas plants with scores from 4 to 9 were considered susceptible [61].

## Inheritance of resistance

Studies of the inheritance of resistance in the CDRK genotype were conducted using 110 RILs derived from the CY population described by [55]. The parental line CDRK is resistant to races 73, 2047, and 3481 of *C. lindemuthianum* and race 63–39 of *P. griseola*, while the parental line Yolano is susceptible to all races.

## Allelism tests

To determine the independence of the ANT resistance allele present in California Dark Red Kidney from *Co-1* and *Co-AC* previously characterized ANT resistance alleles, CDRK was crossed with Andean bean cultivars in the following list: MDRK (*Co-1*) and Amendoim Cavalo (*Co-AC*). In all cases, CDRK was used as the female parent. The $F_1$ seeds were sown in polyethylene vases ($48 \times 30 \times 11$ cm) containing a mixture of the commercial substrate Plantmax®, which had been previously fertilized and sterilized. The plant vases were kept in a greenhouse until the $F_2$ seeds were produced. The $F_2$ individuals, obtained by selfing individual $F_1$ plants, were used to characterize the plants for resistance to race 3481 of *C. lindemuthianum*. Seedlings were grown until pod maturation under natural light in greenhouses. The seedlings were maintained in a greenhouse until the first trifoliolate leaves [56] were fully expanded. Race 3481 of *C. lindemuthianum* was chosen to conduct allelism tests because all parental cultivars inoculated with these races yielded the R × R reaction type.

## Statistical analysis

A goodness-of-fit test for the 1:1 segregation ratio was performed for races 73, 2047, 3481, and 63–39 in the CY RIL population. For allelism tests, segregation analysis of two $F_2$ populations from the crosses of CDRK with Andean cultivars (MDRK and Amendoim Cavalo) was also performed using the $\chi^2$ test according to the Mendelian segregation null hypothesis of 15: 1 R/S ratio.

## SNP genotyping

Total genomic DNA was isolated from the 110 RIL families ($F_{10}$ generation) and parents (CDRK and Yolano) using the DNeasy Plant Mini Kit (Qiagen, CA, USA) following the manufacturer's instructions. The DNA was quantified using 1.5% agarose gel (Agarose SFR, Amresco, IL, USA) with TBE buffer (tris-borate-ethylenediamine tetra acetic acid) and stained with 1 µg mL$^{-1}$ ethidium bromide. The DNA samples were screened with 5,398 SNP DNA markers on the BARCBean6K_3 Illumina BeadChip [63] by following the Infinium HD Assay Ultra Protocol (Illumina, Inc., San Diego, CA, USA). The BeadChip was imaged using the Illumina BeadArray Reader to measure fluorescence intensity. Automatic allele calling for each locus was performed with the Genome Studio Genotyping Module v1.8.4 software (Illumina, San Diego, CA, USA), and all allele calls were visually inspected. Any errors in allele calling due to improper cluster identification were corrected, resulting in 4,633 high-quality SNPs.

## Genome-wide linkage map analysis

SNP markers that were polymorphic between the parents CDRK and Yolano segregated at a 1:1 ratio in the RIL population, as measured by the $\chi^2$ test at p = 0.01, were used to create a linkage map using the default settings of the JoinMap 4.1 software [64]. Briefly, the regression-mapping algorithm based on the Kosambi map function was used to define the linkage order and genetic distances in centiMorgans (cM). A minimum likelihood of odds (LOD) $\geq 3.0$ and a maximum distance of $\leq 50$ cM were used to test linkages among markers. A genetic linkage

map was created using the software MapChart [65]. SNP markers flanking the genomic locations associated with ANT and ALS disease reactions were used to define the physical region of these loci based on the bean reference genome v.1.0 [41] available in NCBI v.1.0 (http://phytozome.jgi.doe.gov).

### Fine-mapping

A fine linkage map was developed with 17 SNPs, two additional SSRs (BARCPVSSR01358, BARCPVSSR01361) and the STS CV542014 markers (http://phaseolusgenes.bioinformatics.ucdavis.edu/markers/1009). The selected SSR and STS markers were amplified from the genomic DNA of the parents CDRK (resistant) and Yolano (susceptible) and 19 of the CY RILs to fine-map a genomic region of 245 Kb between the ss715645251 (50,301,592) and ss715645248 (50,546,985) markers in chromosome Pv01. The primer sequences used to genotype BARCPVSSR01358, BARCPVSSR01361 and STS CV542014 were 'TGGCTGGTTGGTGTTTATGA' (forward) and 'GGTCCCACCCTCTTCTCTTC' (reverse), ´GAATGGTTCATCGTTCATGG´ (forward), and ´TCGGCTGTTTAACGTGGTCT´ (reverse), and `CACTTTCCACTGACGGATTTGAACC` (forward) and `CAGAGGATGCTTCTCACGGT` (reverse), respectively. The PCR mixes contained 30 ng of genomic DNA, 0.25 μM of forward and reverse primers, 1 X PCR Buffer (200 mM Tris-HCl (pH 8.0), 500 mM KCl, 2 mM each dNTP, 10% glycerol, 15 mM $MgCl_2$, and 20 ng/μL of single-strand binding protein (SSB)) and 0.1 unit of Taq DNA polymerase (Invitrogen). The PCR cycle consisted of 3 min at 92˚C; followed by 38 cycles of 50 s at 90˚C, 45 s at 58˚C, and 45 s at 72˚C; a 5 min extension at 72˚C; and a hold at 10˚C. A 2 μL aliquot of loading buffer (30% glycerol and 0.25% bromophenol blue) was added to the DNA products, which were then fractionated on 6% polyacrylamide gels at 3 W $A^{-1}$ $cm^{-1}$. The amplified fragments were stained using SYBR Safe (0.02%), and the DNA bands were visualized under ultraviolet light. Digital images were recorded using an L-PIX Image EX (Loccus Biotecnologia-Loccus do Brasil, Cotia, SP, Brazil).

### Functional annotation of genes linked to ANT and ALS disease reactions

The *P. vulgaris* reference genome v.1.0 [41] was used to define the physical position of the markers flanking the $CoPv01^{CDRK}$/$PhgPv01^{CDRK}$ resistance loci. The putative genes within this genomic region were annotated as candidate genes associated with resistance or susceptibility to ANT and ALS. The putative functional annotation of each candidate gene was based on the descriptions available in Phytozome v.1.0 (https://phytozome.jgi.doe.gov#). After fine-mapping, the new physical region was defined based on the genomic location of the markers flanking the new CDRK resistance loci. Genes predicted within the fine-mapped region were highlighted, and their putative homologs in *Arabidopsis thaliana* were identified using BLASTp in NCBI (National Center for Biotechnology Information; https://www.ncbi.nlm.nih.gov). The *A. thaliana* protein with the lowest E-value (<0.0) and highest identity (>40%) with each bean protein was considered a putative homolog and used to infer its molecular function.

## Results

To identify sources of resistance against the ANT pathogen, a panel of 14 Andean cultivars were screened for their reactions to ten Mesoamerican (9, 64, 65, 73, 89, 445, 453, 1545, 2047, and 3481) and six Andean (2, 7, 19, 23, 39, and 55) races of *C. lindemuthianum* (Table 1). The genomic locations of the known *Co* genes in each cultivar, except for Jalo Vermelho, Jalo Pintado 2, and CDRK (Table 1), have been previously reported [25, 28, 29, 30, 31, 32, 53]. We observed that five genotypes known to carry alleles at the *Co-1* locus (MDRK, Kaboon, Perry Marrow, AND 277, and Widusa) showed different spectra of ANT resistance (Table 1),

**Table 1. Resistant (R) or Susceptible (S) reactions of 15 *Phaseolus vulgaris* cultivars to nine Mesoamerican (9, 65, 73, 89, 445, 453, 1545, 2047, and 3481) and six Andean (2, 7, 19, 23, 39, and 55) races of *C. lindemuthianum* and race 63–39 of *P. griseola*.**

| Cultivar | Genes | *P. griseola* 63–39 | Races of *Colletotrichum lindemuthianum*[a] | | | | | | | | | | | | | | | |
| --- | --- | --- | --- | --- | --- | --- | --- | --- | --- | --- | --- | --- | --- | --- | --- | --- | --- | --- |
| | | | 2 | 7 | 9 | 19 | 23 | 39 | 55 | 64 | 65 | 73 | 89 | 449 | 453 | 1545 | 2047 | 3481 |
| MDRK[b] | *Co-1* | [d]Ne | S | S | R | S | S | S | S | R | R | R | R | R | R | R | S | R |
| Kaboon | *Co-1²* | [d]Ne | R | R | R | R | R | S | S | R | R | R | R | R | R | R | S | R |
| Perry Marrow | *Co-1³* | [d]Ne | R | S | R | R | S | S | S | R | R | R | R | R | S | R | S | R |
| AND 277 | *Co-1⁴* | S | R | S | R | S | R | S | R | R | R | R | R | R | R | R | R | R |
| Widusa | *Co-1⁵* | [d]Ne | R | R | R | S | S | R | S | R | R | R | S | R | R | R | S | S |
| Jalo Vermelho | *Co-12* | R | R | S | R | S | R | S | R | R | R | S | R | R | R | R | S | S |
| Jalo Listras Pretas | *Co-13* | R | S | S | R | S | S | S | S | R | R | R | S | S | S | R | R | R |
| Pitanga | *Co-14* | [d]Ne | R | S | R | R | R | S | R | R | R | S | R | S | S | S | R | S |
| Corinthiano | *Co-15* | R | R | S | S | S | R | [d]Ne | S | R | R | R | R | R | S | R | R | S |
| Paloma | *CoPv01^PA* | [d]Ne | S | S | S | S | R | R | R | R | R | R | S | S | S | R | R | R |
| Amendoim Cavalo | *CoPv01^AC* | R | R | R | R | R | R | R | R | R | R | R | R | R | S | R | R | R |
| Jalo Pintado 2 | *CoPv04^JP2* | R | R | R | R | S | S | S | S | R | R | R | S | R | R | R | R | R |
| Jalo EEP558 | *Co-w, Co-x* | R | S | S | [d]Ne | S | [d]Ne | [d]Ne | S | R | R | R | S | R | R | R | S | R |
| | *Co-y, Co-z* | | | | | | | | | | | | | | | | | |
| CDRK[c] | *CoPv01^CDRK* | R | R | S | R | S | S | R | R | R | R | R | R | S | [d]Ne | R | R | R |
| Yolano [e](MA) | | S | S | S | S | R | S | S | S | S | S | S | S | S | [d]Ne | S | S | S |

[a]Races = Ten Mesoamerican (9, 64, 65, 73, 89, 445, 453, 1545, 2047 and 3481) and six Andean (2, 7, 19, 23, 39, and 55) races of *Colletotrichum lindemuthianum*;

[b]MDRK = Michigan Dark Red Kidney;

[c]CDRK = California Dark Red Kidney;

[d]Ne = Not evaluated;

[e]MA = Mesoamerican.

supporting the hypothesis that they indeed carry different alleles of *Co-1* or additional unidentified *Co* loci. All cultivars except Pitanga were resistant to race 1545 of *C. lindemuthianum*. By contrast, Pitanga and six other genotypes were resistant to race 2047. Kaboon (*Co-1²*), AND 277 (*Co-1⁴*), Jalo Vermelho (*Co-12*), Pitanga (*Co-14*), Corithiano (*Co-15*), and CDRK were the only cultivars that showed resistance and susceptibility to *C. lindemuthianum* races 2 and 7, respectively (Table 1). As CDRK showed resistance to highly virulent races of *C. lindemuthianum* (Table 1) and it is a parent of a well-described RIL population named CY [55], we further characterized the *Co* loci that it might carry. We observed that CDRK exhibited resistance to races 2, 9, 39, 55, 64, 65, 73, 89, 1545, 2047, and 3481 of *C. lindemuthianum*, as well as race 63–39 of *P. griseola*, while Yolano was susceptible to all races (Table 1). Thus, the CY RIL population was used for co-segregation analysis.

## ANT and ALS co-segregation analyses in the CY RIL population

To determine the inheritance of resistance to races 73, 2047, and 3481 of *C. lindemuthianum* and 63–39 of *P. griseola* in CDRK, we inoculated 110 RILs (ten plants from each RIL) with each of these races and scored their disease symptoms (S1 Table). As expected for a single, dominant resistance locus, we observed a segregation ratio of 1RR:1rr in the RIL population, namely, 54 RILs were resistant (RR) and 56 RILs were susceptible (rr) ($\chi^2$ = 0.036, *p-value* = 0.849; Table 2). Interestingly, each RIL showed identical phenotypes in response to each race, indicating that resistances to these ANT and ALS races co-segregate in this population and that the sources of resistance in the CDRK genetic background are either tightly linked. Thus, we named this loci *CoPv01^CDRK*/*PhgPv01^CDRK*.

**Table 2. Segregation for resistance to races 73, 2047, 3481 of *Colletotrichum lindemuthianum* and 63–39 of *Pseudocercospora griseola* in common bean $F_{10}$ RIL population from the California Dark Red Kidney × Yolano cross.**

| Parental cross | Generation | Observed Ratio (1R:1S)[a] | Expected Ratio (1R:1S) | $\chi^2$ | *P* value (1 df) |
|---|---|---|---|---|---|
| Race 73, 2047, and 3481 of *Colletotrichum lindemuthianum* | | | | | |
| CDRK[b] | RP[c] | 30:0 | | | |
| Yolano | SP[d] | 0:30 | | | |
| CDRK × Yolano | $F_{10}$ | 54:56 | 55:55 | 0.036 | 0.849 |
| Race 63–39 of *Pseudocercospora griseola* | | | | | |
| CDRK | RP[b] | 30:0 | | | |
| Yolano | SP[c] | 0:30 | | | |
| CDRK × Yolano | $F_{10}$ | 54:56 | 55:55 | 0.036 | 0.849 |

[a]R = Resistant; S = Susceptible;

[b]CDRK = California Dark Red Kidney;

[c]RP = Resistant Parent;

[d]SP = Susceptible Parent.

## Allelism tests

The results of studies of the allelic relationship between the anthracnose resistance gene in the Andean common bean cultivar CDRK crossed with MDRK and Amendoim Cavalo revealed the absence of allelism (S4 Table). In the allelism test using the $F_2$ population from the cross CDRK × Amendoim Cavalo ($\chi^2 = 0.026$; *p-value* = 0.87) inoculated with race 3481, a segregation ratio of 15R:1S was obtained, indicating the presence of two independent dominant genes; one gene is *Co-AC* [66, 32], present in the cultivar Amendoim Cavalo, and the other gene originated in CDRK. The same ratio of 15R:1S was obtained using race 3481 in the cross CDRK × MDRK ($\chi^2 = 0.022$; *p-value* = 0.88), indicating the action of two dominant genes. In this case, the gene present in CDRK was shown to be independent of the gene *Co-1* [67] present in MDRK.

## Fine-mapping of the *CoPv01^CDRK^/PhgPv01^CDRK^* loci

Genetic linkage analysis between the *CoPv01^CDRK^/PhgPv01^CDRK^* loci and SNPs showing the expected segregation of 1:1 in the RIL population revealed that *CoPv01^CDRK^/PhgPv01^CDRK^* is flanked by the SNP markers ss715645251 and ss715645248 in a genomic region on chromosome Pv01 (Fig 1). The physical locations of the markers ss715645251 and ss715645248 are 50,301,532 bp and 50,546,985 bp, respectively, which correspond to a distance of 245.6 Kb based on the bean reference genome v1.0 (https://www.ncbi.nlm.nih.gov).

To narrow the genomic region harboring the *CoPv01^CDRK^/PhgPv01^CDRK^* loci, we performed a fine-mapping analysis by genotyping 19 RILs that showed recombination events in the 245.6 Kb region identified. Recombination events were identified based on the genotypic data of all 110 RILs obtained with the BARCBEAN6K_3 BeadChip. Upon genotyping these 19 RILs with 12 SNPs, two SSRs, and one STS marker, we observed that the susceptible CY5 RIL and the resistant CY48 RIL contained recombination events (Table 3) that allowed us to delimit the *CoPv01^CDRK^/PhgPv01^CDRK^* region to the area between the CV542014 and ss715645248 markers. Based on the bean reference genome [41], these new *CoPv01^CDRK^/PhgPv01^CDRK^* flanking markers are located at positions 50,513,853 bp (CV542014) and 50,546,985 bp (ss715645248) of chromosome Pv01, spanning 33 Kb (Fig 2).

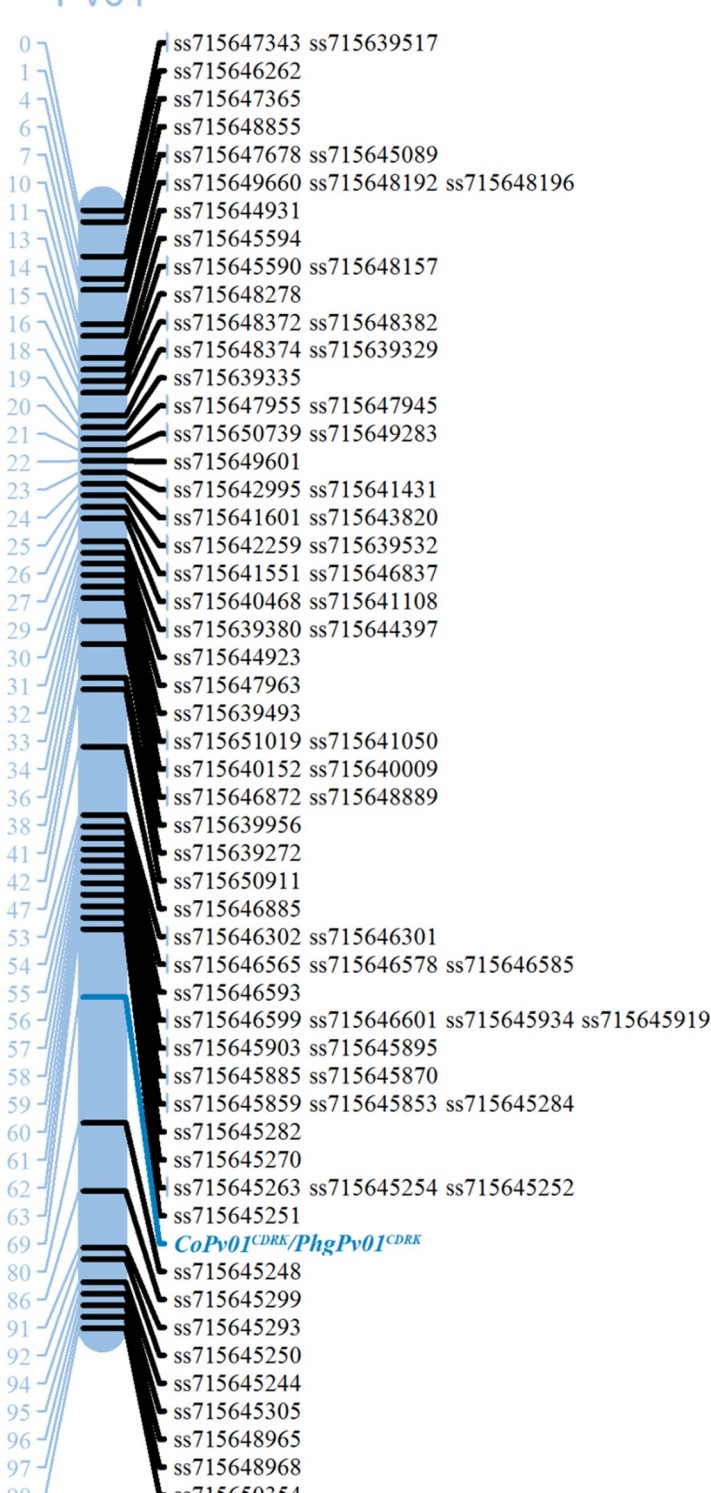

**Fig 1. Genetic map of common bean linkage group Pv01 containing the anthracnose and angular leaf spot resistance loci and linked Single Nucleotide Polymorphism (SNPs) markers used to genotype the $F_{10}$ population California Dark Red Kidney × Yolano.** Recombination distances are indicated on the left side of the linkage group in centiMorgans (cM), and the marker names are shown on the right side. The $CoPv01^{CDRK}/PhgPv01^{CDRK}$ resistance loci were flanked by SNP markers ss715645251 and ss715645248 in $F_{10}$ mapping population. The map was drawn with MapChart (65).

**Table 3. Genotype and phenotype of 19 $F_{10}$ recombinant events in the region of Pv01 used for fine mapping of the anthracnose and angular leaf spot resistance loci in CDRK.** The phenotype was obtained from the reaction of the 110 $F_{10}$ RILs to races 73, 2047, and 3481 of *Colletotrichum lindemuthianum* and race 63–39 of *P. griseola*. Genotyping was achieved using the flanking markers 12 SNP, two SSR and one STS markers that enabled the positioning of the $CoPv01^{CDRK}/PhgPv01^{CDRK}$ loci in a 33 kb genomic region flanked by markers CV542014 and ss715645248.

| Marker | SNP position | Recombinant lines from CDRK × Yolano | | | | | | | | | | | | | | | | | | |
|---|---|---|---|---|---|---|---|---|---|---|---|---|---|---|---|---|---|---|---|---|
| | | 5 | 12 | 19 | 20 | 33 | 38 | 43 | 47 | 48 | 62 | 70 | 73 | 79 | 87 | 88 | 91 | 115 | 96 | 146 |
| ss715645260 | 50115685 | AA | AA | BB | BB | AA | BB | AA | AA | AA | BB | AA | AA | AA | BB | BB | AA | BB | AB | AB |
| ss715645259 | 50130201 | AA | AA | BB | BB | AA | BB | AA | AA | AA | BB | AA | AA | AA | BB | BB | AA | BB | AB | AB |
| ss715645258 | 50155987 | AA | AA | BB | BB | AA | BB | AA | AA | AA | BB | AA | AA | AA | BB | BB | AA | BB | AB | AB |
| ss715645257 | 50161526 | AA | AA | BB | BB | AA | BB | AA | AA | AA | BB | AA | AA | AA | BB | BB | AA | BB | AB | AB |
| ss715645256 | 50182775 | AA | AA | BB | BB | AA | BB | AA | AA | AA | BB | AA | AA | AA | BB | BB | AA | BB | AB | AB |
| ss715645254 | 50203547 | AA | AA | BB | BB | AA | BB | AA | AA | AA | BB | AA | AA | AA | BB | BB | AA | BB | AB | AB |
| ss715645252 | 50222584 | AA | AA | BB | BB | AA | BB | AA | AA | AA | BB | AA | AA | BB | BB | BB | AA | BB | AB | AB |
| ss715645251 | 50301592 | AA | AA | BB | BB | AA | BB | AA | AA | AA | BB | AA | AA | BB | BB | BB | AA | BB | AB | AB |
| BARCPVSSR01358 | 50350345 | AA | AA | BB | BB | AA | BB | - | AA | AA | BB | AA | AA | BB | BB | BB | AA | BB | BB | AA |
| BARCPVSSR01361 | 50388017 | AA | AA | BB | BB | AA | BB | - | AA | AA | BB | AA | AA | BB | BB | BB | AA | BB | BB | AA |
| **CV542014** | **50513853** | AA | AA | BB | BB | AA | BB | AA | AA | AA | BB | AA | AA | BB | BB | BB | AA | BB | AB | AB |
| $CoPv01^{CDRK}/PhgPv01^{CDRK}$ | | **BB** | **AA** | **BB** | **AA** | **BB** | **BB** | **AA** | **AA** | **AA** | **BB** | **BB** | **AA** | **BB** | **BB** | **AA** | **BB** | **BB** | **BB** | **AA** |
| **ss715645248** | **50546985** | BB | AA | BB | BB | AA | BB | AA | AA | BB | BB | AA | AA | BB | BB | BB | AA | BB | AB | AB |
| ss715645299 | 51353193 | BB | BB | AA | BB | AA | AA | AA | BB | BB | BB | BB | BB | BB | BB | BB | BB | AA | AB | AA |
| ss715645293 | 51617802 | BB | BB | AA | AA | BB | AA | BB | BB | BB | AA | AA | BB | BB | AA | AA | BB | AA | AA | AA |
| ss715645250 | 51726047 | BB | BB | AA | AA | BB | AA | BB | BB | BB | AA | AA | BB | BB | AA | AA | BB | AA | BB | AA |
| ss715645244 | 51764167 | AA | BB | AA | AA | BB | AA | BB | BB | BB | AA | AA | BB | BB | AA | AA | BB | AA | BB | AA |
| ss715645305 | 51786948 | AA | BB | AA | AA | BB | AA | BB | BB | BB | AA | AA | BB | BB | AA | AA | BB | AA | BB | AA |
| ss715645301 | 51819821 | AA | BB | AA | AA | BB | AA | BB | BB | BB | AA | AA | BB | BB | AA | AA | BB | AA | BB | AA |
| ss715648967 | 51883712 | AA | BB | AA | AA | BB | AA | BB | BB | BB | AA | AA | BB | BB | AA | AA | BB | AA | BB | AA |
| ss715648965 | 51896315 | AA | BB | AA | AA | BB | AA | BB | BB | BB | AA | AA | BB | BB | AA | AA | BB | AA | BB | AA |

AA = Resistant; BB = Susceptible; AB = Heterozygous;— = not available.

## Predicted genes and functions associated with the $CoPv01^{CDRK}/PhgPv01^{CDRK}$ loci

Based on the bean reference genome, the 33 Kb region contains five predicted genes (Table 4). The predicted genes encode the proteins Phvul_001G246000 (ATP-dependent RNA Helicase), Phvul.001G246100 (Cation-dependent mannose-6-phosphate receptor), Phvul.001G246200 (Protein Trichome Birefringence-like 33), Phvul.001G246300 (Abscisic Acid (ABA) receptor PYL5), and Phvul.001G246400 (SNF2 domain-containing protein classy 1-related). The putative orthologs for these genes were identified in the Arabidopsis genome (TAIR) for further functional reference (Table 4).

Interestingly, the marker CV542014 is physically mapped at 946 bases upstream the stop codon of the Phvul.001G246000, at the last predicted intron (Fig 2). Furthermore, the marker ss715645248 is located at 1,283 bases from the stop codon of the predicted gene Phvul.001G246400 (Fig 2), resulting in a mutation in the last exon of this gene, which encodes for a putative DEXH-box Helicase Domain (DEXHc_ATRX-like; Conserved Domain cd18007, e-value = $9.8 \times 10^{-73}$).

Additionally, the predicted gene Phvul.001G245300, which encodes a putative Leucine-Rich Repeat Protein Kinase (LRR-Kinase), was detected approximately 66 Kb upstream of the marker locus CV542014 (Table 4). An additional putative LRR-kinase (Phvul.001G246800)

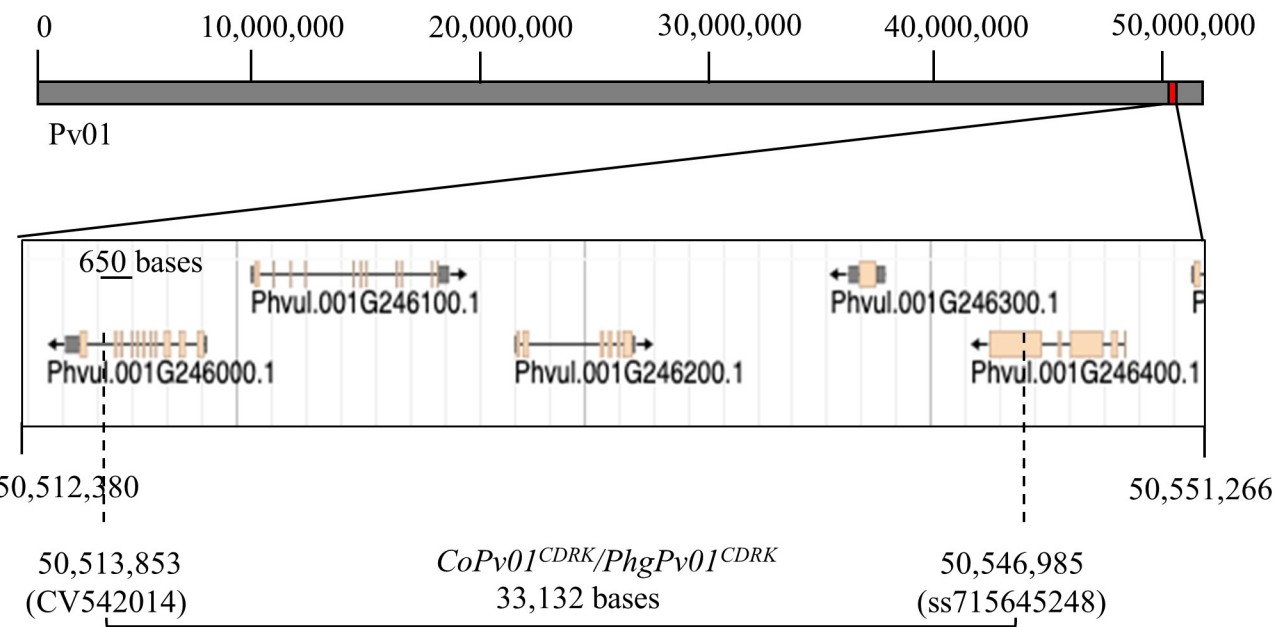

**Fig 2. Fine mapped region for the CDRK resistance loci*CoPv01^CDRK^/PhgPv01^CDRK^*.** Upper bar represents the entire chromosome Pv01, in which the *CoPv01^CDRK^/PhgPv01^CDRK^* region is highlighted as red square at the end of the chromosome. The five predicted genes within this region are shown, where the location of the *CoPv01^CDRK^/PhgPv01^CDRK^* flanking markers CV542014 and ss715645248 are indicated by dashed lines, within the predicted genes Phvul.001G246000 and Phvul.001G246400, respectively. The genomic region between these markers is indicated by the lower bar and cover around 33 Kbp of the genome.

was found 26 Kb downstream of the marker ss715645248 (Table 4). These five genes are interesting candidates for *CoPv01^CDRK^/PhgPv01^CDRK^* and may confer resistance to ANT and ALS pathogens. The Fig 3 shows the ANT resistance cluster present at the end of chromosome Pv01.

**Table 4. Gene models found in delimitated region CDRK resistance loci against anthracnose and angular leaf spot were fine-mapped, gene position in reference genome v1.0 and annotation.**

| Gene Model in *P. vulgaris* | Homolog in *A. thaliana* | E-value[a] | Identity[a] | Functional annotation on TAIR[b] | Functional annotation on Phytozome[c] |
|---|---|---|---|---|---|
| Phvul.001G245300 | AT4G18640 | $8 \times 10^{-169}$ | 43.40% | Male Discoverer 2 | Protein tyrosine kinase (pkinase_tyr) // leucine rich repeat n-terminal domain (lrrnt_2) |
| Phvul.001G246000 | AT5G05450 | 0 | 70.50% | RNA Helicase 18 | ATP-dependent RNA helicase ddx55/spb4 [ec:3.6.4.13] (ddx55, spb4) |
| Phvul.001G246100 | AT2G37390 | $1 \times 10^{-105}$ | 52.60% | Sodium Potassium Root Defective 2 | Cation-dependent mannose-6-phosphate receptor |
| Phvul.001G246200 | AT2G40320 | 0 | 69.90% | Trichome Birefringence-Like 33 | Protein trichome birefringence-like 33 |
| Phvul.001G246300 | AT2G40330 | $1 \times 10^{-76}$ | 56.90% | ABA Receptor PYL6 | Abscisic acid receptor pyl5 |
| Phvul.001G246400 | AT5G20400 | 0 | 48.30% | Flavanone 3 Hydroxylase-like | SNF2 domain-containing protein classy 1-related |
| Phvul.001G246800 | AT3G50740 | 0 | 60.60% | UDP-Glucosyl Transferase 72E1 | Leucine-rich repeat receptor-like protein kinase imk3-related |

[a]E-values and Identity for BLASTp analysis performed on NCBI (National Center for Biotechnology Information; https://www.ncbi.nlm.nih.gov)

[b] Functional gene annotation resource: TAIR—The Arabidopsis Information Resource (https://www.arabidopsis.org)

[c]Functional gene annotation resource: Phytozome—Common bean reference genome v1.0 (https://phytozome.jgi.doe.gov#

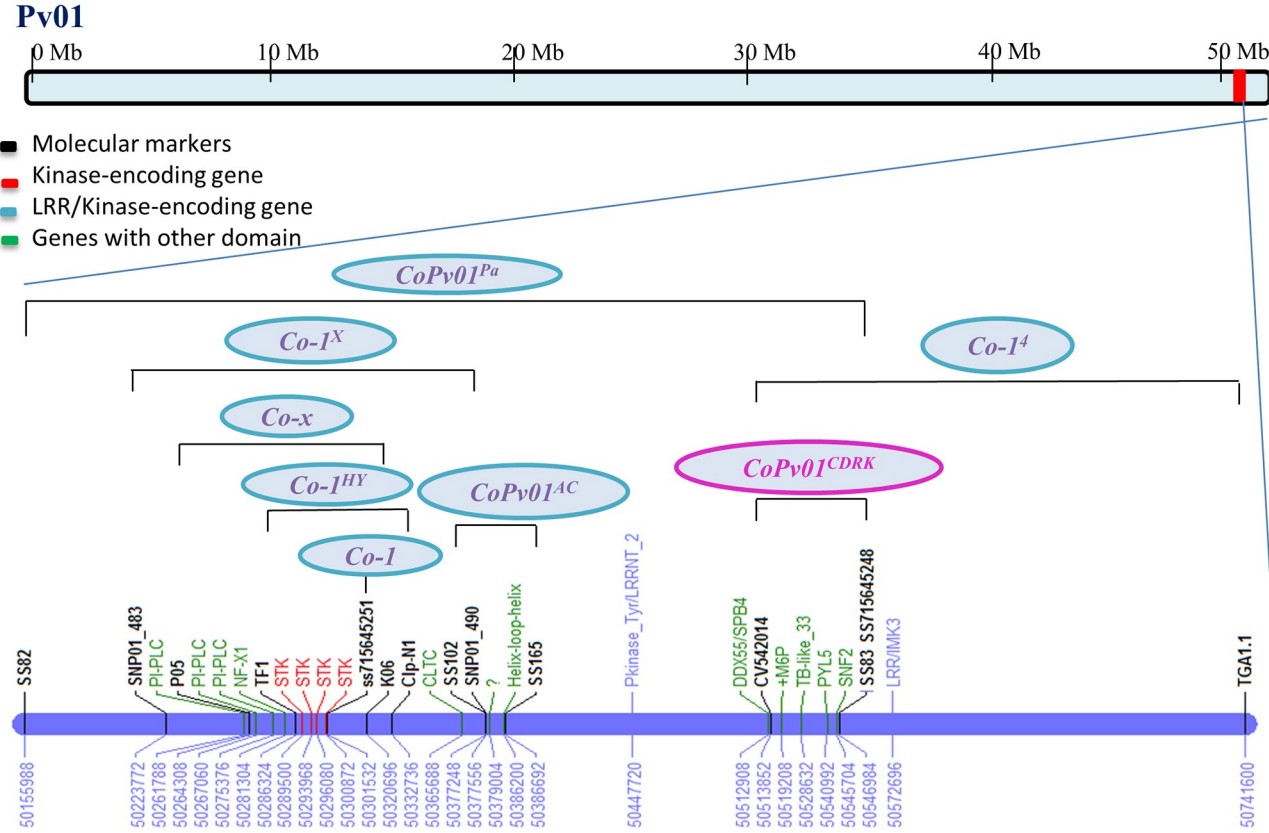

**Fig 3. Anthracnose cluster 1.1 on Pv01 with specific candidate genes within and close to the genomic region where these resistance genes were mapped.** Molecular markers linked to the resistance genes are displayed in black color. In red candidate genes that encode kinases, in blue candidate genes encoding NB-LRR and in green candidate genes with other domains. The resistance gene in CDRK cultivar was fine mapped in a region having five candidate genes and close to LRR. *Co-1^HY^*, *Co-x* and *Co-1^X^* harbor the same kinases. *CoPv01^AC^*, *Co-1^4^* and *CoPv01^PA^* are the resistance genes of the Amendoim Cavalo, AND 277 and Paloma cultivars, respectively.

## Discussion

The spectra of resistance observed for CDRK were different from those of the four Andean cultivars that have ANT resistance alleles at the *Co-1* locus. CDRK is resistant to race 2 while MDRK is susceptible. In relation to races 39, 55 and 2047 CDRK was resistant and MDRK, Kaboon and Perry Marrow were susceptible. Additionally, AND 277 showed compatible reaction to race 39. The resistance of CDRK to races 2, 9, 64, 65, and 73 of *C. lindemuthianum* and race 63–39 of *P. griseola* is important for common bean breeding programs in Brazil, where these races have high rates of occurrence [45, 46, 47, 49].

The results of inoculating the 110 CY RILs with races 73, 2047, and 3481 of *C. lindemuthianum* and 63–39 of *P. griseola* showed that 54 RILs were resistant, while 56 were susceptible. These results fit a segregation ratio of 1R:1S, revealing a monogenic inheritance. It is important to note that all RILs resistant to race 73 were also resistant to races 2047, 3481, and 63–39, while all RILs susceptible to race 73 were also susceptible to races 2047, 3481, and 63–39.

The present study established that the resistance of CDRK to races 73, 2047, and 3481 of *C. lindemuthianum* and race 63–39 of *P. griseola* is conferred by a dominant gene. The dominant nature of resistance in the CDRK cultivar suggests that resistance is transferable to commercial cultivars.

In addition, allelism tests conducted in two $F_2$ populations from crosses R × R between the CDRK × MDRK, and CDRK × Amendoim Cavalo inoculated with race 3481 revealed the presence of two dominant genes in each population that conferred resistance to anthracnose. One of these genes (*Co-1*) is present in MDRK [67], and the other gene is present in CDRK. Notably, the segregation results for the $F_2$ population from the cross CDRK × Amendoim Cavalo highlighted the presence of a dominant gene in CDRK that is independent of the *Co-AC* genes present in Amendoim Cavalo. The monogenic nature of Amendoim Cavalo resistance to *C. lindemuthianum* race 3481 was reported by [31]. These results support the hypothesis that the gene conferring resistance to race 3481 of this fungal pathogen, present in CDRK, is independent from other genes (*Co-1* and *Co-AC* genes), harbored in Michigan Dark Red Kidney and Amendoim Cavalo, respectively.

In this study, we elucidated the inheritance of anthracnose resistance in the Andean bean cultivar CDRK and established the genetic relationship between this resistance locus and two known ANT resistance genes mapped to Cluster 1.1 at the end of Pv01. Additional segregation analysis in populations derived from crossing CDRK with two Andean bean cultivars, each carrying different ANT resistance genes, revealed that the ANT resistance of CDRK is conferred by a new gene that is distinct from the previously reported resistance genes *Co-1* and *Co-AC* in common bean. Based on this evidence, the authors propose that the symbol for anthracnose resistance loci in the CDRK cultivar be named *CoPv01*$^{CDRK}$/*PhgPv01*$^{CDRK}$.

The co-segregation analysis of the ANT and ALS resistance genes was observed in 110 CY RILs originating from a CDRK × Yolano cross, which showed identical phenotypes in response to both diseases. A total of 54 RILs that were resistant to the ANT pathogen were also resistant to all races of the ALS pathogen; 56 RILs that were susceptible (S) to the ANT pathogen were also susceptible to the ALS pathogen. These results suggest that the *CoPv01*$^{CDRK}$ and *PhgPv01*$^{CDRK}$ loci are closely linked.

ANT and ALS are two of the most widespread and severe diseases of common bean in the Americas and Africa, which are considered the two largest producers and consumers of this crop [8, 11]. Moreover, these genes protect common bean against two pathogens possessing high and possibly rapidly changing virulence diversity. Thus, the availability of accurate molecular markers to transfer these genes into commercial common bean cultivars would probably increase resistance durability in these cultivars against highly variable pathogens. In the present study, we conducted a co-segregation analysis of the *CoPv01*$^{CDRK}$ and *PhgPv01*$^{CDRK}$ loci using two sets of CY RILs, which were inoculated independently with specific races of the ANT and ALS pathogens, to ensure accurate phenotypic evaluations. We combined separate co-segregation tests for ALS and ANT using a large set of phenotyped plants (4,400 $F_{10}$ plants) with the power of an Illumina BeadChip (containing over 5,398 SNPs). These large sets of information enabled the creation of a genetic linkage map and revealed the genetic distance between the genes *CoPv01*$^{CDRK}$ and *PhgPv01*$^{CDRK}$ at the end of chromosome Pv01. Most of the disease-resistance genes of common bean that have already been characterized are located in distal chromosome regions with high levels of recombination, thus favoring the identification of molecular markers closely linked to disease-resistance genes. However, the presence of repeated sequences, such as khipu satellites, at several large resistance clusters of Pv04, Pv10, and Pv11 might complicate the process of designing locus-specific primers [32].

The candidate region containing the *CoPv01*$^{CDRK}$ and *PhgPv01*$^{CDRK}$ loci in linkage group Pv01 is located close to one of the most important clusters of ANT resistance genes in the common bean genome [32]. The Pv01 cluster includes the following ANT disease-resistance genes: *Co-1*, *Co-1*$^{HY}$, *Co-1*$^4$; *Co-x*, *Co-Pa*, *Co-AC*, and *Co-1*$^X$ [23, 25, 28, 29, 30, 31, 53, 68]. The *CoPv01*$^{CDRK}$/*PhgPv01*$^{CDRK}$ loci was found in a genomic region flanked by the markers ss715645251 (50,301,532 bp) and ss715645248 (50,546,925 bp) on Pv01 (Fig 1).

Furthermore, the kinase Phvul.001G243800 was significantly associated with responses to races 65, 73, and 3481 in a genome-wide association study of Andean bean lines [68]. This study used an Andean diversity panel (ADP) and identified the SNP marker ss715645251, which was associated with ANT, in the gene Phvul.001G243800 at position 50,301,532; the authors attributed this gene to the *Co-1* locus [68]. Moreover, the $Co1^{HY}$ resistance gene in the Hongyundou cultivar was fine-mapped between the markers TF1 and Clp-N1, positioned at 50,286,325 and 50,332,737, respectively. This 46 Kb region harbors four candidate genes: Phvul.001G243500, Phvul.001G243600, Phvul.001G243700, and the previously mentioned candidate gene for the *Co-1* locus, Phvul.001G243800 [29].

The resistance gene *Co-x* was identified in the Jalo EEP558 cultivar and fine-mapped to a genomic region (56 Kb) flanked by the markers P05 and K06 at positions 50,264,307 and 50,320,695. This 56 Kb region overlaps the 34 Kb of $Co-1^{HY}$, which contains the four candidate genes mentioned above, and the larger region harbors three additional genes, Phvul.001G243200, Phvul.001G243300, and Phvul.001G243400 [28]. The resistance gene in the Xana cultivar, named $Co-1^{X}$, was mapped between the markers SNP01_483 and SNP01_490, located in a 153 Kb region from 50,223,771 and 50,377,556. A total of 17 candidate genes were identified in this region, which also covers part of the $Co-1^{HY}$ and *Co-x* region [53].

Furthermore, the resistance gene of Amendoim Cavalo was recently fine-mapped between the markers SS102 and SS165, located at 50,377,247 and 50,386,692. This 9 Kb interval harbors three candidate genes: Phvul.001G244300, Phvul.001G244400, and Phvul.001G244500. The first of these genes is also present in the $Co-1^{X}$ interval, sharing 0.309 kb.

The AND 277 resistance allele ($Co-1^{4}$) was mapped between the markers CV542014[450] (50,513,853) at 0.7 cM and TGA1.1[570] (50,741,598) at 1.3 cM. Fine-mapping studies are being conducted to reduce this 227 Kb interval. Mapping analysis located the $CoPv01^{PA}$ resistance gene in a 390 Kb region flanked by the SNP markers SS82 (50,155,987) and SS83 (50,546,985) at distances of 1.3 and 2.1 cM, respectively. This region contains 46 annotated genes, nine of which contain domains with functions related to pathogen resistance: Phvul.001G243200, Phvul.001G243300, Phvul.001G243500, Phvul.001G243600, Phvul.001G243700, the above-mentioned Phvul.001G243800, Phvul.001G243100, Phvul.001G245100 and Phvul.001G245300.

$CoPv01^{CDRK}$ was fine-mapped to a 33 Kb interval between CV542014 (50,513,853) and ss715645248 (50,546,985) harboring Phvul.001G246000, Phvul.001G246100, Phvul.001G246200, Phvul.001G246300 and Phvul.001G246400 and close to the genes Phvul.001G245300 and Phvul.001G246800 that encode NB-LRR domains. In this context and based on the physical positions of the aforementioned markers, $CoPv01^{CDRK}$ is clearly positioned downstream of the *Co-1*, $Co-1^{HY}$, *Co-x*, $CoPv01^{AC}$, and $Co-1^{X}$ loci (Fig 3). These results suggest that the $CoPv01^{CDRK}$ locus is different from the aforementioned loci.

Among the candidate genes identified in the $CoPv01^{CDRK}$/$PhgPv01^{CDRK}$ loci, the genomic sequences of Phvul.001G246000 and Phvul.001G246400 contain the locus-flanking markers CV542014 and ss715645248, respectively (Fig 2, Table 4). Phvul.001G246000 is a homolog of *A. thaliana* RNA Helicase 18 (RH18), which was linked to spontaneous chlorosis in hybrids [69]. Several members of the RNA helicase family were found to be involved in the chlorotic phenotype in young leaves by affecting chloroplast biogenesis and reducing photosynthesis [70, 71]. Interestingly, *C. lindemuthianum* was previously shown to modulate the expression of several genes predicted to be located in the chloroplast [33], and a decrease in plant photosynthetic rates was observed after ANT infection [72]. Indeed, chlorosis can arise due to a delay in chloroplast biogenesis, functionality, or metabolism [73]. Therefore, Phvul.001G246000 could affect photosynthesis to control the chlorosis induced by ANT and ALS, causing resistant

plants not to present disease symptoms. Similarly, Phvul.001G246400 is a homolog of Arabidopsis Flavanone 3 Hydroxylase-like (F3H-like), a flavonol synthase involved in the accumulation of flavonoids under light [74]. Hence, it is possible that Phvul.001G246400 could be involved in the biosynthesis of flavonoids to avoid light stress, preventing damage in the chloroplast and contributing to the absence of necrosis after fungal infection. Moreover, flavonoids are secondary metabolites known to be induced upon pathogenic attack, mainly by the induction of the phytohormone jasmonic acid (JA) [75]. JA is well known as an antifungal plant hormone and has previously been linked to defense against ANT in common bean [33] and ALS [36]. The $CoPv01^{CDRK}$/$PhgPv01^{CDRK}$ loci also contains Phvul.001G246300, a homolog of Arabidopsis ABA Receptor PYL6 (Table 4), which plays a central role in the crosstalk between the ABA and JA responses [76]. This suggests that Phvul.001G246300 may have a role in JA/ABA responses during common bean interaction with *C. lindemuthianum* and *P. griseola*.

Moreover, Phvul.001G246100 in the $CoPv01^{CDRK}$/$PhgPv01^{CDRK}$ loci is a homolog of Arabidopsis Sodium Potassium Root Defective 2 (NAKR2), which belongs to a family of proteins known to have a heavy-metal-associated domain that is linked to the cell division rate in the root meristem [77]. Another gene in the $CoPv01^{CDRK}$/$PhgPv01^{CDRK}$ loci is Phvul.001G246200, a homolog of Arabidopsis Trichome Birefringence-Like 33 (TBL33), a protein known to be involved in the synthesis and deposition of secondary wall cellulose [78]. Finally, receptor-like kinases are known to function in plant defense against pathogens [79], including common bean response to ANT and ALS [33, 36]. Therefore, we also investigated the closest LRR-Kinases to the fine-mapped $CoPv01^{CDRK}$/$PhgPv01^{CDRK}$ loci (Table 4). Phvul.001G245300 was shown to be a homolog of Arabidopsis Male Discoverer 2 (MDIS2), a receptor located in the pollen tube that perceives a female attractant signal to promote fertilization [80]. Phvul.001G246800 is homologous to Arabidopsis UDP-Glucosyl Transferase 72E1 (UGT72E1), an enzyme thought to be involved in lignin metabolism [81]. Although none of these proteins has any directly established function in plant defense against pathogens, it is possible that NAKR2, TBL33, and UGT72E1 have roles in plant cell wall strengthening to avoid fungal penetration. MDIS2 is involved in plant fertilization [80]. Another protein putatively involved in plant fertility and linked to ANT resistance in beans is Feronia-like, which is closely related to the ANT resistance gene *COK-4* [54] and is thought to function in the regulation of both plant growth and defense [82]. Therefore, it seems that proteins involved in pollen–gametophyte recognition have roles in common bean defense against ANT and possibly also against ALS.

## Conclusions

The results presented here showed that CDRK cultivar has the $CoPv01^{CDRK}$ and $PhgPv01^{CDRK}$ genes conferring resistance to races 73, 2047, and 3481 of *C. lindemuthianum* and race 63–39 of *P. griseola*. The $CoPv01^{CDRK}$ and $PhgPv01^{CDRK}$ loci co-segregated and were completely linked on chromosome Pv01. New resistance loci, $CoPv01^{CDRK}$ and $PhgPv01^{CDRK}$, against *C. lindemuthianum* and *P. griseola* were fine-mapped in a genomic region of 33 Kb on chromosome Pv01 that harbors five predicted genes. Allelism tests showed that $CoPv01^{CDRK}$ resistance gene is different from the *Co-1* and *Co-AC* loci mapped on Pv01; the physical distances of these genes from $CoPv01^{CDRK}$ are 211,376 bp and 126,216 bp, respectively. In addition, $CoPv01^{CDRK}$ and $PhgPv01^{CDRK}$ resistance alleles are inherited together and can be monitored with high efficiency using SNP markers. These results can be applied to breeding programs aimed at developing bean cultivars with ANT and ALS resistance using marker-assisted selection.

## Supporting information

**S1 Fig. Reaction of the $F_2$ seedlings derived from the cross California Dark Red Kidney (CDRK) × Michigan Dark Red Kidney (MDRK) inoculated with race 3481 of *C. lindemuthianum* for allelism tests.**
(TIF)

**S1 Table. Disease reaction (resistance = R or susceptibility = S) in 110 $F_{10}$ RILs (CY = California Dark Red Kidney × Yolano population) to races 73, 2047 and 3481 of *C. lindemuthianum* and race 63–39 of *P. griseola*.** Ten plants per each RIL were evaluated. Lines written in italics underlined carry recombinant events in the predicted location of the $Co^{CDRK}$/$Phg^{CDRK}$ loci.
(DOC)

**S2 Table. Gene models found in the Pv01 region between 50,301,592 and 50,301,592 delimited by the markers ss715645251 and ss715645248, respectively.** Gene positions and their functional annotations are based on the bean reference genome available at Phytozome.
(DOC)

**S3 Table. SNP markers associated with the anthracnose resistance locus in the common bean cultivar CDRK discovered by co-segregation and genetic mapping analysis and located on the lower end of chromosome Pv01 of common bean.** *Genetic position in centi-Morgans (cM), based on the map developed by Song et al. (2015).
(DOC)

**S4 Table. Allelism tests in $F_2$ populations for the anthracnose resistance gene in the common bean cultivar California Dark Red Kidney (CDRK).**
(DOC)

## Acknowledgments

M.C. Gonçalves-Vidigal is grateful for grant from Capes. The authors acknowledge Dr Paul Gepts of the University of California Davis, USA, for kindly provided the seeds of CY RILS, and Dr Pedrina Gonçalves Vidigal for critical reading of the manuscript.

## Author Contributions

**Conceptualization:** M. Vaz-Bisneta, P. S. Vidigal Filho.

**Data curation:** G. Valentini, M. Vaz-Bisneta, P. S. Vidigal Filho.

**Formal analysis:** M. C. Gonçalves-Vidigal, T. A. S. Gilio, M. Vaz-Bisneta.

**Funding acquisition:** P. S. Vidigal Filho, Q. Song.

**Investigation:** M. C. Gonçalves-Vidigal, G. Valentini, M. Vaz-Bisneta, P. S. Vidigal Filho.

**Methodology:** M. C. Gonçalves-Vidigal, T. A. S. Gilio, G. Valentini, M. Vaz-Bisneta, Q. Song, P. R. Oblessuc.

**Project administration:** M. C. Gonçalves-Vidigal.

**Supervision:** Q. Song, M. Melotto.

**Visualization:** M. Vaz-Bisneta, P. S. Vidigal Filho.

**Writing – original draft:** M. C. Gonçalves-Vidigal, P. R. Oblessuc, M. Melotto.

**Writing – review & editing:** M. C. Gonçalves-Vidigal.

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
