## [Decision Letter · Decision Letter 0]

11 Mar 2020

PONE-D-20-00761

New Andean source of resistance to anthracnose and angular leaf spot: fine-mapping of disease-resistance genes in California Dark Red Kidney common bean cultivar

PLOS ONE

Dear Dr. Maria Celeste Gonçalves-Vidigal,

Thank you for submitting your manuscript to PLOS ONE. After careful consideration, we feel that it has merit but does not fully meet PLOS ONE’s publication criteria as it currently stands. Therefore, we invite you to submit a revised version of the manuscript that addresses the points raised during the review process.

ACADEMIC EDITOR: 

The manuscript was reviewed by two reviewers and both recommended the manuscript after addressing some minor changes/comments. Therefore we advise authors to incorporate these minor changes suggested by both the reviewers and resubmit the manuscript. 

We would appreciate receiving your revised manuscript by %March 30, 2020%. To enhance the reproducibility of your results, we recommend that if applicable you deposit your laboratory protocols in protocols.io, where a protocol can be assigned its own identifier (DOI) such that it can be cited independently in the future. For instructions see: http://journals.plos.org/plosone/s/submission-guidelines#loc-laboratory-protocols

We look forward to receiving your revised manuscript.

Kind regards,

Reyazul Rouf Mir, PhD

Academic Editor

PLOS ONE

Additional Editor Comments (if provided):

The manuscript entitled "New Andean source of resistance to anthracnose and angular leaf spot: fine-mapping of disease-resistance genes in California Dark Red Kidney common bean cultivar" is dealing with most important disease in common bean. The manuscript reports the fine mapping of disease-resistance genes in California Dark Red Kidney common bean cultivar. The information will prove useful in gene cloning and use of genes in marker-assisted selection for breeding common bean cultivars with enhanced disease resistance. Based on the advise received by two reviewers, I recommend acceptance of manuscript after minor revision. The authors may consider minor comments made by the two reviewers.

Journal Requirements:

"M.C. Gonçalves-Vidigal is grateful for grant from Capes number BEX 88881.170662//2018-01."

3. Your ethics statement must appear in the Methods section of your manuscript. If your ethics statement is written in any section besides the Methods, please move it to the Methods section and delete it from any other section. Please also ensure that your ethics statement is included in your manuscript, as the ethics section of your online submission will not be published alongside your manuscript.

Reviewers' comments:

Reviewer's Responses to Questions

**Comments to the Author**

1. Is the manuscript technically sound, and do the data support the conclusions?

Reviewer #1: Yes

Reviewer #2: Yes

2. Has the statistical analysis been performed appropriately and rigorously? 

Reviewer #1: Yes

Reviewer #2: Yes

3. Have the authors made all data underlying the findings in their manuscript fully available?

Reviewer #1: Yes

Reviewer #2: Yes

4. Is the manuscript presented in an intelligible fashion and written in standard English?

Reviewer #1: Yes

Reviewer #2: Yes

5. Review Comments to the Author

Reviewer #1: The manuscript entitled "New Andean source of resistance to anthracnose and angular leaf spot: fine-mapping of disease-resistance genes in California Dark Red Kidney common bean cultivar" by Gonçalves-Vidigal et al is an interesting subject and very well written. Anthracnose and angular leaf spot caused by Colletotrichum lindemuthianum and Pseudocercospora griseola , respectively, are devastating diseases of common bean around the world. The have concluded with the results by showing the fine mapping of CoPv01 CDRK / PhgPv01 CDRK loci to a smaller genomic region of 33 Kb. This information can be applied in breeding programs to develop bean cultivars with ANT and ALS resistance using marker-assisted selection.

Some minor comments for authors to address arguing below

Line no. 128: Please mention number of plants evaluated for non-inoculated control

Line no. 148: Elaboration is required for how and where F1 hybrids seeds were selfed to obtain F2 i.e whether the procedure development of F2 from F1 was done in field conditions or in control conditions

Line no. 78: Uniform style citing the reference

While citing a reference in the introduction section, authors are suggested to cite only the original reference and furthermore reduce the references cited for a particular sentence. Eg in line No. 73, there are eleven references quoted by the author, better is to reduce it as minimum

In the introduction and discussion sections, support your conclusions regarding fine mapping with recent published reports of 2020.

Lastly authors are requested to please go through the author guidelines of the journal before submitting the final revised manuscript.

Reviewer #2: The manuscript is well written and the information at Co-1 locus is well described. The new R gene and it mapping will assist development of breeding lines in the region. I therefore recommend its publication. There are few typos that need attention.

L124 please mention authors

L124 please check Inocula

L128 rewrite

L135 by [61, 62]. please add authors

L135 Between change it to from

L191 space after and" and , before and

6. PLOS authors have the option to publish the peer review history of their article (what does this mean?). If published, this will include your full peer review and any attached files.

Reviewer #1: No

Reviewer #2: Yes: Bilal A Padder

---

## [Author Response · Author response to Decision Letter 0]

1 Apr 2020

DATE: March 28, 2020

TO: Reyazul Rouf Mir, PhD, Academic Editor PLOS ONE

FROM: Maria Celeste Gonçalves-Vidigal, Corresponding Author

SUBJECT: Manuscript [PONE-D-20-00761] - [EMID:c34843a9dba2cb7e]

We appreciate your thoughtful comments and suggestions to improve the manuscript entitled "New Andean source of resistance to anthracnose and angular leaf spot - fine-mapping of disease-resistance genes in California Dark Red Kidney common bean cultivar" based on the recommendations on the cover letter. 

I received this communication from you on a e-mail dated Wednesday, March 11, 2020 - onbehalfof+ PloS One Editorial Office:em@editorialmanager.com

 We have carefully reviewed the manuscript and believe that this problem has been properly addressed in the revised manuscript. The revisions and corrections are shown in red colored font on the text of the revised manuscript and all response to the reviewer comments are provided this letter.

Thank you for your collaboration and attention.

Comments from the Editor and Reviewers to the Authors

Reviewer #1

 'The manuscript entitled "New Andean source of resistance to anthracnose and angular leaf spot: fine-mapping of disease-resistance genes in California Dark Red Kidney common bean cultivar" by Gonçalves-Vidigal et al isan interesting subject and very well written. Anthracnose and angular leaf spot caused by Colletotrichum lindemuthianum and Pseudocercospora griseola, respectively, are devastating diseases of common bean around the world. They have concluded with the results by showing the fine mapping of CoPv01CDRK / PhgPv01CDRKloci to a smaller genomicregion of 33 Kb. This information can be applied in breeding programs to develop bean cultivars with ANT and ALS resistance using marker-assisted selection.

Some minor comments for authors to address arguing below:'

Line no. 128: Please mention number of plants evaluated for non-inoculated control.

Page # 6, Lines 131 to 138: All plants were inoculated and the paragraph has been rewritten: 

'After the expansion of the first trifoliolate leaf, the leaflet of 10 seedlings of each line were inoculated with each race of C. lindemuthianum and the leaflet of 10 seedlings with P. griseola. Each pathogen was inoculated separately. A spore suspension containing 2.0 × 106 spores mL-1 of races 73, 2047 and 3481 of C. lindemuthianum were inoculated using a DeVilbiss number 15 atomizer powered by an electric air compressor (Schulz, SA, Joinville, Santa Catarina, Brazil). A similar procedure was employed for the inoculation with race 63-39 of P. griseola using a spore suspension of 1.2 × 104 conidia mL-1. Ten plants for resistant and susceptible control for each race were inoculated [59, 60].'

Line no. 148: Elaboration is required for how and where F1 hybrids seeds were selfed to obtain F2 i.e whether the procedure development of F2 from F1 was done in field conditions or in control conditions

Page #7, Lines # 156 to 162

The paragraph starting at the line 148 to 149, the paragraph has been rewritten: 

'The F1 seeds were sown in polyethylene vases (48 × 30 × 11 cm) containing a mixture of the commercial substrate Plantmax®, which had been previously fertilized and sterilized. The plant vases were kept in a greenhouse until the F2 seeds were produced. The F2 individuals, obtained by selfing individual F1 plants, were used to characterize the plants for resistance to race 3481 of C. lindemuthianum. Seedlings were grown until pod maturation under natural light in greenhouses. The seedlings were maintained in a greenhouse until the first trifoliolate leaves [56] were fully expanded.'

Line no. 78: Uniform style citing the reference.

While citing a reference in the introduction section, authors are suggested to cite only the original reference and furthermore reduce the references cited for a particular sentence. Eg in line No. 73, there are eleven references quoted by the author, better is to reduce it as minimum.

Pages # 3 and 4, Lines # 70 to 75: The number of authors in the line # 73 was reduced and they were added along of the paragraph according to the specific phrase of the author, the paragraph has been rewritten as see below :

'Resistance to ANT is conferred by independently segregating individual loci in a series named Co. Currently, the known Co genes are Co-1 and its alleles, Co-14, Co-Pa, Co-x, and Co-w on chromosome Pv01 [22, 23, 25, 27, 28, 30, 31, 32]; Co-u and CoPv02 on chromosome Pv02 [18, 22]; Co-13 and Co-17 on chromosome Pv03 [29]; Co-3, Co-32, Co-33, Co-34/Phg-3, Co-y, Co-z, and Co-RVI on chromosome Pv04 [24, 26]; Co-5, Co-6, and Co-v on chromosome Pv07; and Co-2 on chromosome Pv11 [29].'

In the introduction and discussion sections, support your conclusions regarding fine mapping with recent published reports of 2020.

Page # 29, Lines # 634 to 637: A new bibliographic reference was added in the Introduction on the mapping of anthracnose resistance genes in Pv01. Consequently, it was also mentioned in the Discussion

'Farooq M, Padder BA, Bhat NN, Shah MD, Shikari AB, Awale HE, Kelly JD. Temporal expression of candidate genes at the Co-1 locus and their interaction with other defense related genes in common bean. Physiol Mol Plant Pathol. 2019; 108: 101424. doi: 10.1016/j.pmpp.2019.101424.'

Lastly authors are requested to please go through the author guidelines of the journal before submitting the final revised manuscript.

The guidelines was checked and changes were made 

Reviewer #2

The manuscript is well written and the information at Co-1 locus is well described. The new R gene and it mapping will assist development of breeding lines in the region. I therefore recommend its publication. There are few typos that need attention.

L124 please mention authors

Page # 6, Line # 126: The authors were mentioned: 

'Monosporic cultures of C. lindemuthianum and of P. griseola were prepared according to the methodologies proposed by Mathur et al. [57] and Sanglard et al. [58].'

L124 please check Inocula

Page # 6, Line #127: The word Inocula was replaced by Inoculum

L128 rewrite

Page # 6, Lines # 130 to 137, the paragraph has been rewritten as see below:

'Soon after the expansion of the first trifoliolate leaf, the leaflet of 10 seedlings of each line were inoculated with each race of C. lindemuthianum and the leaflet of 10 seedlings with P. griseola. Each pathogen was inoculated separately. A spore suspension containing 2.0 × 106 spores mL-1 of races 73, 2047 and 3481 of C. lindemuthianum were inoculated using a DeVilbiss number 15 atomizer powered by an electric air compressor (Schulz, SA, Joinville, Santa Catarina, Brazil). A similar procedure was employed for the inoculation with race 63-39 of P. griseola using a spore suspension of 1.2 × 104 conidia mL-1. Ten plants for resistant and susceptible control for each race were inoculated [59, 60].'

 L135 by [61, 62].please add authors

L135 Between change it to from

Page # 6, Line 142: The name of authors were added and the word 'from' was inserted:

Pastor-Corrales et al. [61] and Inglis et al. [62]. Plants with disease reaction scores from 1 

L191 space after and" and , before and

Page # 9, Line # 205: The space was inserted 

Funding: This research was supported by Brazilian Federal Funding Institutions National Council for Scientific and Technological Development (CNPq) for financial support and scholarship grants and the Coordination for the Improvement of Higher Education Personnel (Capes). M.C. Gonçalves-Vidigal is grateful for grant from Capes number BEX 88881.170662//2018-01.

We would like to thank for the suggestions from the Editor and the reviewers that have improved this manuscript.

Regards,

M. C. Gonçalves-Vidigal

Corresponding Author

---

## [Decision Letter · Decision Letter 1]

11 Jun 2020

New Andean source of resistance to anthracnose and angular leaf spot: fine-mapping of disease-resistance genes in California Dark Red Kidney common bean cultivar

PONE-D-20-00761R1

Dear Dr. Maria Celeste Gonçalves-Vidigal,

We’re pleased to inform you that your manuscript has been judged scientifically suitable for publication and will be formally accepted for publication once it meets all outstanding technical requirements.

Kind regards,

Reyazul Rouf Mir, PhD

Academic Editor

PLOS ONE

Additional Editor Comments (optional):

Reviewers' comments:

Reviewer's Responses to Questions

**Comments to the Author**

1. If the authors have adequately addressed your comments raised in a previous round of review and you feel that this manuscript is now acceptable for publication, you may indicate that here to bypass the “Comments to the Author” section, enter your conflict of interest statement in the “Confidential to Editor” section, and submit your "Accept" recommendation.

Reviewer #1: All comments have been addressed

Reviewer #2: All comments have been addressed

2. Is the manuscript technically sound, and do the data support the conclusions?

Reviewer #1: Yes

Reviewer #2: Yes

3. Has the statistical analysis been performed appropriately and rigorously? 

Reviewer #1: Yes

Reviewer #2: Yes

4. Have the authors made all data underlying the findings in their manuscript fully available?

Reviewer #1: Yes

Reviewer #2: Yes

5. Is the manuscript presented in an intelligible fashion and written in standard English?

Reviewer #1: Yes

Reviewer #2: Yes

6. Review Comments to the Author

Reviewer #1: All the queries and suggestions pointed out in the manuscript "New Andean source of resistance to anthracnose and angular leaf spot: fine-mapping of disease-resistance genes in California Dark Red Kidney common bean cultivar" have been well addressed and presented in the revised manuscript

Reviewer #2: I have no further queries but a suggestion. Please check reference no 23, Farooq et al., After checking it I found Mahiya-Farooq on running head of the paper. So in my opinion it should be Mahiya-Farooq.

7. PLOS authors have the option to publish the peer review history of their article (what does this mean?). If published, this will include your full peer review and any attached files.

Reviewer #1: No

Reviewer #2: No

---

## [Editor Report · Acceptance letter]

18 Jun 2020

PONE-D-20-00761R1 

New Andean source of resistance to anthracnose and angular leaf spot: fine-mapping of disease-resistance genes in California Dark Red Kidney common bean cultivar 

Dear Dr. Gonçalves-Vidigal:

I'm pleased to inform you that your manuscript has been deemed suitable for publication in PLOS ONE. Congratulations! Your manuscript is now with our production department. 

Kind regards, 

on behalf of

Dr. Reyazul Rouf Mir 

Academic Editor

PLOS ONE